# Air quality co-benefits for human health and agriculture counterbalance costs to meet Paris Agreement pledges

Toon Vandyck[1], Kimon Keramidas [1], Alban Kitous[1], Joseph V. Spadaro[2], Rita Van Dingenen[3], Mike Holland[4] & Bert Saveyn[1]

Local air quality co-benefits can provide complementary support for ambitious climate action and can enable progress on related Sustainable Development Goals. Here we show that the transformation of the energy system implied by the emission reduction pledges brought forward in the context of the Paris Agreement on climate change (Nationally Determined Contributions or NDCs) substantially reduces local air pollution across the globe. The NDCs could avoid between 71 and 99 thousand premature deaths annually in 2030 compared to a reference case, depending on the stringency of direct air pollution controls. A more ambitious 2 °C-compatible pathway raises the number of avoided premature deaths from air pollution to 178–346 thousand annually in 2030, and up to 0.7–1.5 million in the year 2050. Air quality co-benefits on morbidity, mortality, and agriculture could globally offset the costs of climate policy. An integrated policy perspective is needed to maximise benefits for climate and health.

[1] European Commission, Joint Research Centre (JRC), 41092 Sevilla, Spain. [2] Spadaro Environmental Research Consultants (SERC), 19142 Philadelphia, USA. [3] European Commission, Joint Research Centre (JRC), 21027 Ispra, Italy. [4] Ecometrics Research and Consulting (EMRC), RG8 7PW Reading, UK. Correspondence and requests for materials should be addressed to T.V. (email: toon.vandyck@ec.europa.eu)

Causing 5.6–6.6 million premature deaths in 2016[1], air pollution is an important externality that interacts with climate change. Researchers[2–5] have called for a coordinated effort to combat climate change and to improve air quality, but little work has been done to analyse the global synergies between the Paris Agreement on climate change[6] and air pollution. This paper combines extensive data sets and models on emissions, climate, the energy system, the dispersion and impacts of ambient air pollutants, and the economy to quantify the impact of actual climate change mitigation policies proposed in the run-up to the 21st Conference Of the Parties in Paris on three inter-related Sustainable Development Goals[7]: Good health (SDG3), Clean energy (SDG7), and Climate action (SDG13). As emphasised by the Intergovernmental Panel on Climate Change (IPCC)[8], a comprehensive analysis of co-benefits and adverse side effects is essential to estimate the actual costs of mitigation policies. In a political context, the co-benefits on air pollution are particularly relevant because they are mainly local and short term, while the averted climate change impacts occur globally over a decadal temporal scale.

While earlier regional estimates present a broad range of values for the co-benefits of climate policy on air quality (between 2 and 196$ per tonne of carbon dioxide, with mean value of 49$/tCO$_2$)[9], recent work[10–12] highlights that the improved human health outcomes due to cleaner air can largely offset the costs to reduce greenhouse gases (GHGs), particularly in heavily polluted regions. Here we assess the global and regional mortality, morbidity, and agricultural air quality co-benefits in the context of the Paris Agreement while accounting for future uncertainty in air pollution control measures. The main contribution lies in the quantification of ancillary benefits of actual (pledged) climate and energy policy elements in the Nationally Determined Contributions (NDCs), thereby capturing the heterogeneity in country-specific ambition levels and in sector coverage, in contrast with scenarios studying global and economy-wide carbon pricing, and complementing recent studies for the US[13–15] and China[16–19]. Considering the regional differentiation in pledges is crucial because the transboundary effects of air pollution can be

substantial[20]. The results show that the NDCs as pledged in the run-up to the Paris Agreement could improve health outcomes substantially, avoiding 71–99 thousand premature deaths in the year 2030. Present study furthermore compares the value of the air quality co-benefits with the macroeconomic cost of climate change mitigation policies, the latter also depending on the relative ambition levels across countries through industry competitiveness and international trade. We find that the value of co-benefits differs widely across regions and outweighs the costs of reducing GHGs on a global level in the majority of scenarios.

## Results

**Climate change mitigation pathways and the energy system.** The climate scenarios encompass three trajectories of GHG emissions (global aggregate shown in Fig. 1; details by region in Supplementary Table 1). The Reference (REF) assumes no climate change mitigation policies beyond those already in place and compares best to Current policy scenarios in the literature[21]. The NDC scenario implements the GHG emission reductions and related policies in the Paris pledges, including the emission reductions that are conditional on other aspects of the Paris Agreement such as financing. Jointly, the current pledges represented by this scenario imply a likely increase in global average temperature of 2.5–3.2 °C, in line with the range provided by other studies[21]. The 2 °C scenario considers policies that result in a trajectory of GHG emissions that is consistent with at least 75% probability of limiting the average rise of global temperature to 2 °C by 2100 compared to pre-industrial levels.

Air quality policies are likely to develop in parallel of climate policies and may affect the potential scope of co-benefits of climate policy[22]. We address the implications of air pollution policy uncertainty for the co-benefits by exploring three storylines. The Fixed Legislation (FLE) scenario considers no additional implementation of air pollution abatement technologies from 2010 onwards. In combination with a REF climate policy, this assumption implies that economic and population growth lead to increasing global emissions over time for all air

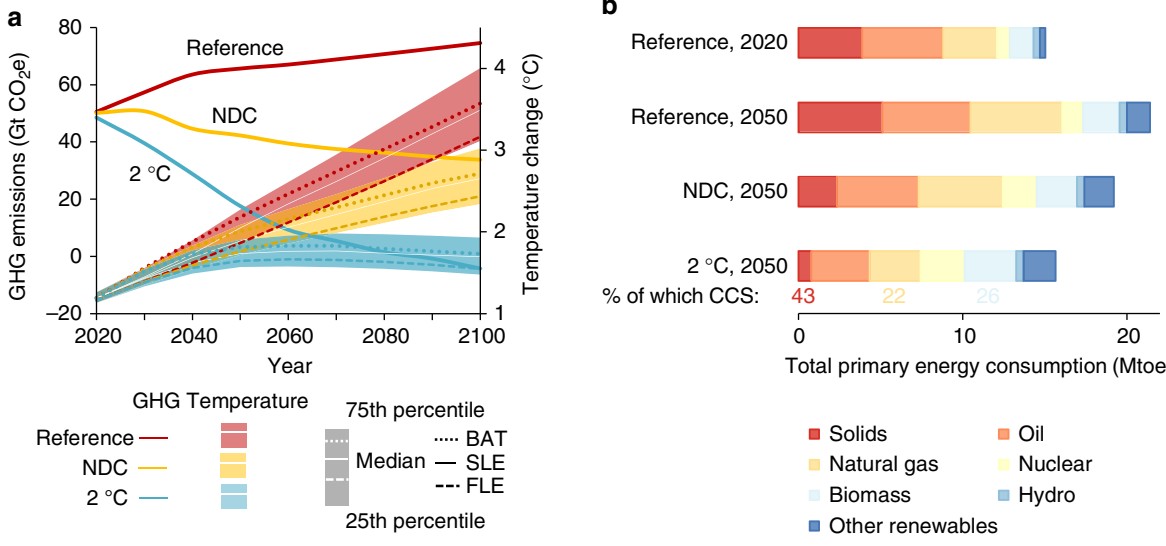

**Fig. 1** Climate policy scenarios and corresponding energy systems. The figure shows greenhouse gas emissions, global average temperature change compared to pre-industrial levels, and energy consumption by source in the three climate policy scenarios. **a** Greenhouse gas emissions and median temperature changes over the course of the century. Median temperature changes in the case of Best Available Technologies (BAT) and Fixed Legislation (FLE) air quality scenarios are shown by dotted lines above and below the white line indicating the Stringent Legislation (SLE), respectively. Shaded area indicates 25th and 75th percentiles. **b** Total primary energy consumption by source. Other renewables include solar, wind, and geothermal energy. The percentage of solids, gas, and biomass energy consumption that uses carbon capture and storage (CCS) is indicated for the 2 °C scenario

pollutants, with the exception of carbon monoxide (CO), for which a rising trend is offset by ongoing progress in energy technology and corresponding efficiency improvements. Because high levels of air pollution provide a broad base for reductions, the estimates of the co-benefits of climate policy derived under the assumption of Fixed air quality Legislation will be considered here as an upper bound. A gradual adoption and diffusion of air pollution control measures is included in the Stringent Legislation (SLE) scenario, which better reflects ambitious recent policy objectives in fast-growing countries, such as China. In the Best Available Technologies (BAT) scenario, countries fully adopt the maximum technically feasible air pollutant emission reduction technologies by 2030. This hypothetical benchmark identifies how structural changes induced by climate policy can improve air quality beyond what can be expected by end-of-pipe air pollution abatement technologies alone. By implementing stringent air pollution abatement, the BAT scenario leaves less room for co-benefits of climate action and will be used here to quantify a lower bound for the co-benefits.

Air pollutants affect global and local temperature changes and regional precipitation patterns[23,24]. The effect of air pollution control on climate change is a priori unclear, because the radiative forcing of some pollutants, such as black carbon (BC), is positive, while other pollutants ($NO_x$, $SO_2$) have a cooling effect on the climate. In addition to the median estimate of global average temperature change and the 50% probability bounds for the SLE scenario, Fig. 1 (dotted lines) includes the median estimate in case of less (FLE) and more (BAT) stringent air pollution control measures. The figure shows that the temperature change relative to pre-industrial levels in the REF climate policy under more ambitious air pollution controls (BAT) exceeds the central case estimate (SLE) by 0.08 °C in 2100, while the higher end of the air pollution projection (FLE) implies temperature changes that are 0.32 °C below the SLE case in 2100. Hence, the end-of-pipe reduction of air pollutants with a cooling effect outweighs the decrease in air pollutants that contribute to global warming in our scenarios, leading to a net upward effect on global mean temperatures, in line with other work[25,26]. This result does not consider the effect of air pollution controls on GHG emissions and is less pronounced in more ambitious climate mitigation scenarios. Compared to the REF, the decrease in GHG emissions in the 2 °C scenario implies larger reductions in global mean temperature in the case of cleaner air (1.84 °C under BAT vs. 1.62 °C under FLE, REF—2 °C in 2100), since co-reduction of cooling aerosols plays a smaller role when stringent air pollution controls (BAT) are in place.

Ambitious climate policies are to reshape the energy landscape in the coming decades[27–29]. The decarbonisation of the energy supply mix combined with reduced energy consumption through efficiency gains will be key factors in the transformation of energy systems (Fig. 1; regional numbers are given in Supplementary Tables 2–5). Energy efficiency drives total global energy consumption down by approximately 10% in the NDC scenario and by more than a quarter (27%) in the 2 °C scenario in the year 2050 compared to the REF. Moreover, the rising share of renewables represents a structural change in the energy sector, especially in electricity generation.

**The impact of climate policy measures on air quality.** Policies that aim to mitigate climate change tend to reduce emissions of GHGs and local air pollutants[22,30], particularly when both share the same underlying drivers. The extent to which $CO_2$ reductions are correlated with changes in air pollutants differs by region and type of air pollutant (Fig. 2). For pollutants that mainly result

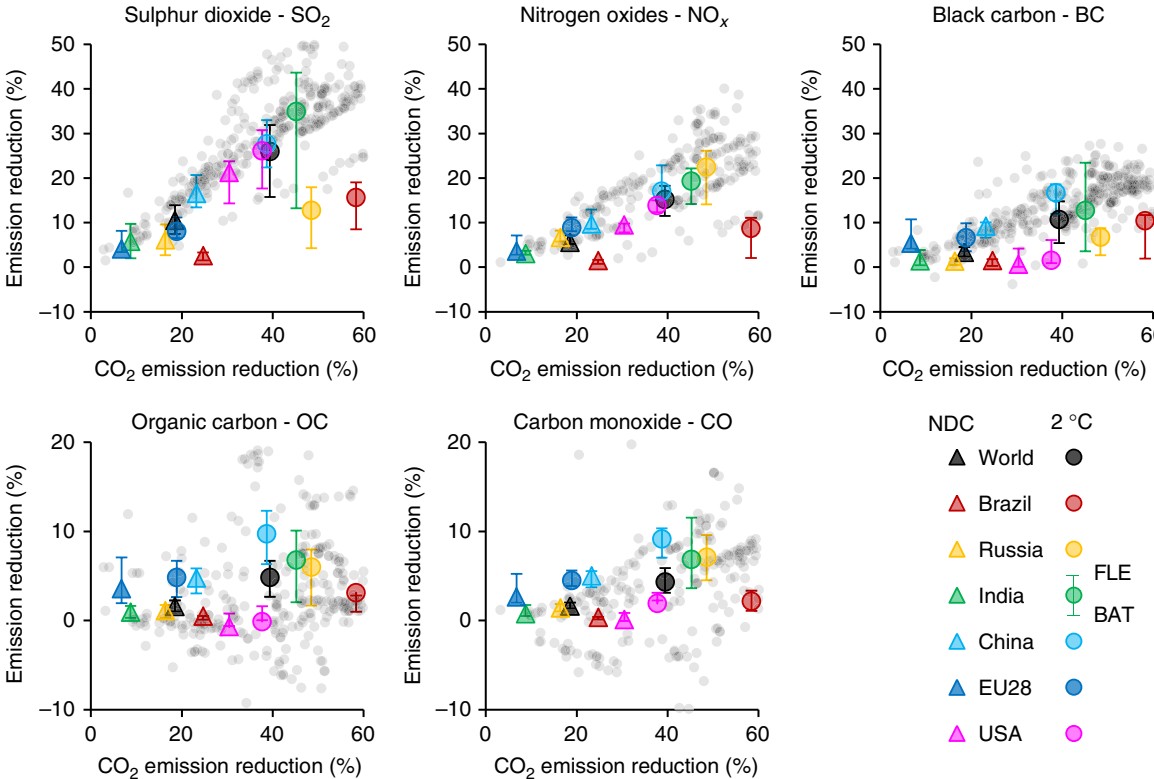

**Fig. 2** Co-movement of emissions of $CO_2$ and air pollutants per region due to climate change mitigation policies. Emission reductions are expressed as percentage difference from the respective Reference climate scenario emissions cumulative over 2015–2050. Symbols represent the results for the Stringent Legislation air quality scenario, while the whiskers indicate the results for the Fixed Legislation (FLE) and Best Available Technologies (BAT) air quality scenarios. Grey dots show global results obtained from IPCC AR5 WGIII[35]

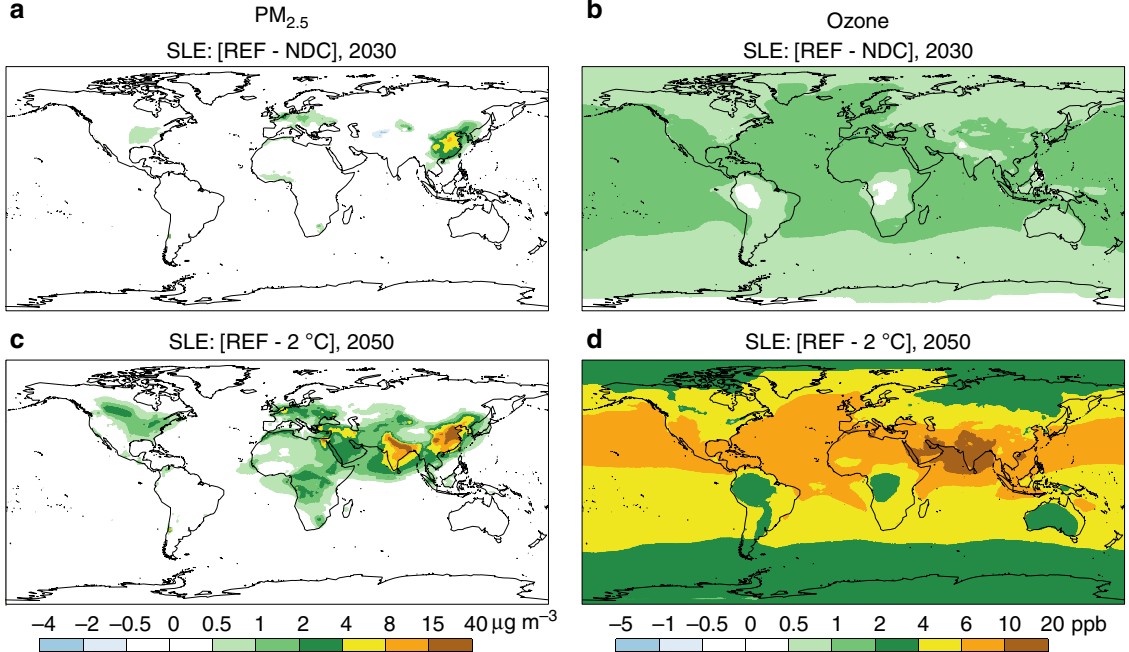

**Fig. 3** Reduction of $PM_{2.5}$ concentration ($\mu g\ m^{-3}$) and ozone mixing ratio (ppb) due to climate change mitigation policies under the Stringent Legislation case of air quality. NDC scenario compared with the Reference in 2030 (REF—NDC) for **a** $PM_{2.5}$ and **b** ozone. 2 °C scenario compared with the Reference in 2050 (REF—2 °C) for **c** $PM_{2.5}$ and **d** ozone. Positive values indicate improved air quality

from the combustion of fossil fuels, such as $SO_2$ and $NO_x$, emission reductions tend to be strongly correlated with decreasing emissions of $CO_2$. This co-movement is less obvious for regions with large GHG abatement potential from options other than the burning of fossil fuels, such as land use in Brazil (see Supplementary Table 6), and for regions with important industrial sources of pollutants, such as $SO_2$ from production of metals in Russia. Figure 2 shows that air quality co-benefits generally outweigh adverse side effects on the aggregate level for most pollutants and regions, although there are some trade-offs embedded in technological choices such as carbon capture and storage[31,32], biomass energy[33], and biofuels[34]. For organic carbon and CO in particular, Fig. 2 shows that not all models in the IPCC's Fifth Assessment Report[35] agree on the sign of the change in emissions on the global level, indicating uncertainty in the estimates. Figure 2 furthermore indicates the sensitivity of co-reductions with respect to the implemented air pollution control technologies (Supplementary Tables 7–28 provide numbers by region and scenario). Low air pollutant emission intensities (BAT) in the benchmark in key climate change mitigation sectors reduce the scope for co-benefits, as can be seen from the sulphur dioxide emission reductions in India, for instance, where the implementation of BAT implies less polluting coal-fired electricity generation facilities.

The impacts of air pollution are not confined to national borders as air pollutants are dispersed geographically[20,36]. The changes in the concentration of particulate matter with diameter smaller than 2.5 μm ($PM_{2.5}$) and tropospheric ozone mixing ratio mapped in Fig. 3 are the results of emissions, transportation, and atmospheric chemistry reactions of pollutants (see detailed numbers in Supplementary Tables 29–34). Hence, results for PM include both direct emissions from primary sources, such as BC and organic matter, and secondary PM that derives from emissions of $NH_3$, $NO_x$, $SO_2$, and volatile organic compounds (VOCs). Ozone is formed by the reaction of precursor gases $NO_x$, VOCs, and CO in the presence of sunlight. By 2030, the time horizon of most NDCs, the Paris pledges lead to globally

relatively small but locally significant reductions in the concentration of $PM_{2.5}$, while a decrease in ozone mixing ratio spreads more widely across the globe. Over the long run (2050), the impact of a more ambitious climate policy setting (2 °C) reveals that the potential contribution of GHG abatement policies to improved air quality is substantial, particularly in China, India, and the Middle East, where benchmark concentrations are comparably high.

**Air quality co-benefits for human health and agriculture**. The benefits of improved air quality include avoided premature mortality due to related cardiovascular and respiratory diseases and lung cancer[37]. Consistent with other research[38], outdoor air pollution-related premature mortality is projected to roughly double by 2050 compared to 2010 when considering the current climate (REF) and air pollution (FLE) policies combined with population and economic growth, whereas ambitious policies (2 °C—BAT) bring the number of premature deaths below the level of 2010 despite population growth by 2050 (premature mortality per scenario, region, and pollutant are detailed in Supplementary Tables 35–40). Climate policies as currently pledged under the NDCs lead to between 71 and 99 thousand avoided premature deaths globally in the year 2030 compared to current climate policies (REF), while bringing greenhouse emissions in line with a 2 °C temperature goal prevents between 178 and 346 thousand premature deaths globally in the year 2030. In the year 2050, 2 °C-compatible climate action reduces the premature deaths from air pollution by 0.7–1.5 million compared to the REF, of which more than two thirds are prevented in India and China (Fig. 4). The gap between the NDC and 2 °C scenarios is explained by countries like India, for which the current NDC up to 2030 fails to capture the potential air quality co-benefits of climate action. In addition to avoided premature mortality, reductions in air pollution bring benefits in terms of reduced sickness days, thereby boosting labour markets (see Supplementary Table 41).

Ground-level ozone penetrates leaves and hinders plant growth, thereby affecting agricultural productivity[39,40]. By

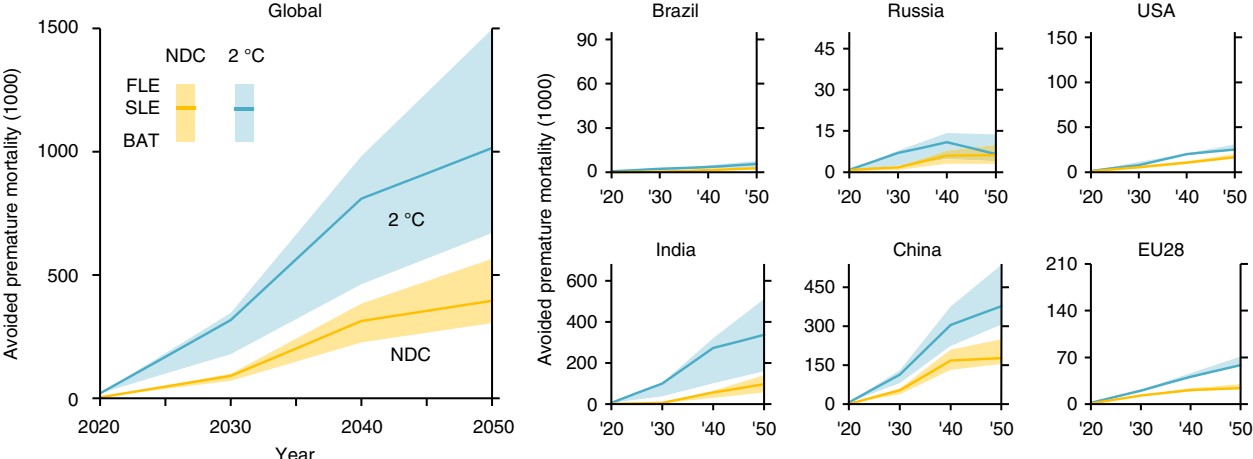

**Fig. 4** Avoided premature mortality due to lower PM$_{2.5}$ concentration and ozone mixing ratio implied by climate change mitigation policies. Results shown are relative to the corresponding climate policy Reference case and use non-linear exposure-response functions based on Global Burden of Disease 2015[25], with the range encompassing estimates with Fixed Legislation (FLE), Stringent Legislation (SLE), and Best Available Technologies (BAT) air quality scenarios. In addition to global results, the figure shows avoided premature mortality (in thousands) for six regions. To display the absolute numbers in a way that makes the regional figures comparable against each other, the range of the vertical axes is scaled to 0.04% of the region-specific population in 2050

reducing ozone precursor emissions, global climate policy can improve crop yields. Using exposure–response functions (ERFs) for seven crop types, we calculate the crop yield impact of climate policy-induced reductions in tropospheric ozone mixing ratios. Projecting future changes in crop yield onto current values of agricultural output, we obtain the impacts shown in Fig. 5, where monetary agricultural co-benefits are expressed in per capita terms. The results for the NDC scenario highlight the areas where ozone reductions overlap with high production values of ozone-sensitive crops, such as maize, soybeans, and wheat in the US or sugar cane and soybeans in Brazil. Whereas the NDCs globally raise the yields of maize, rice, soy, and wheat by 0.4–0.7%, 0.1–0.3%, 0.8–1.1%, and 0.4–0.6% in 2030, respectively, more ambitious climate policy limiting global warming to 2 °C increases productivity of those crops by 0.8–1.5%, 0.2–0.8%, 1.8–2.7%, and 0.9–1.7% compared to the REF in 2030 (see also Supplementary Table 42). Estimates of monetary agricultural co-benefits corresponding to yield impacts for the year 2050 exceed 10$ per capita in some regions, when calculated using current-day value of agricultural production. Importantly, increased agricultural crop yields could contribute to reaching the Sustainable Development Goal No hunger (SDG2).

**Comparing climate policy's costs and co-benefits.** The air quality co-benefits of the NDC and 2 °C-consistent climate policies on avoided premature mortality, reduced lost work days due to sickness, and improved agricultural crop yields more than offset the cost of climate change mitigation policies on a global average over the 2015–2050 period under the majority of scenario settings (Fig. 6). This result implies that air quality provides further justification to GHG abatement policies, in addition to avoided climate change damages, which are not considered here. Whereas crop yields and work days are reflected by a (agriculture and labour) market value, the avoided premature mortality is largely a nonmarket co-benefit, evaluated here by using the Value of Statistical Life (VSL). The low, medium, and high VSLs are heterogeneous across regions and grow with income over time (Supplementary Table 43), as empirical studies typically find that the willingness-to-pay for health risk reductions varies with income. For comparison, we also include the results with a VSL

that is homogeneous across regions and fixed in time at a more conservative value. Compared to an approach based on years of life lost, the use of the same VSL for all deaths may bias the valuation upwards when the population affected is characterised by relatively old age, poor health conditions, and comparably short life expectancy.

The value of air quality co-benefits per tonne of GHGs abated differs substantially across regions. Relatively high values for China confirm earlier findings[10,12,41], while high numbers for India point to the potential domestic gains if the country were to step up the level of climate policy ambition. The results for Europe reveal the transboundary benefits of the Paris Agreement on a reduction of ozone mixing ratio worldwide. As relatively ambitious climate policies are already adopted in the EU (hence included in the REF), the additional GHG reductions to reach the NDC target are relatively small, while the region's air quality improves also due to climate change mitigation efforts in other regions. A high population density, relatively clean air in the REF (which places the starting point for the health analysis in the steep part of the non-linear ERF), strong energy efficiency improvements in the residential and transport sectors, and an important component of secondary PM (for which reductions tend to correlate well with $CO_2$ emission reductions, see $SO_2$ and $NO_x$ in Fig. 2) in overall particulate composition[42] further contribute to the co-benefits in Europe. The bottom–up nature of the Paris pledges results in a wide range of mitigation costs across countries. Results presented in Fig. 6 account for the impact on competitiveness on international markets and incorporate the macroeconomic cost of decreasing global demand for fossil fuel-producing countries' exports. The impact on crop yields (Supplementary Table 44) accounts for the growth of the agricultural sector over time, in contrast to the results presented in Fig. 5. Induced technological change is not included, which may lead to the cost estimate to be biased upwards. In order to attribute the impacts uniquely to climate policy, we derive the costs (as well as the co-benefits) of climate policy by comparing the NDC and 2 °C scenario with the REF under the same assumption for air pollution control (e.g. REF—FLE vs. NDC—FLE) and therefore exclude the costs of air pollution control technologies.

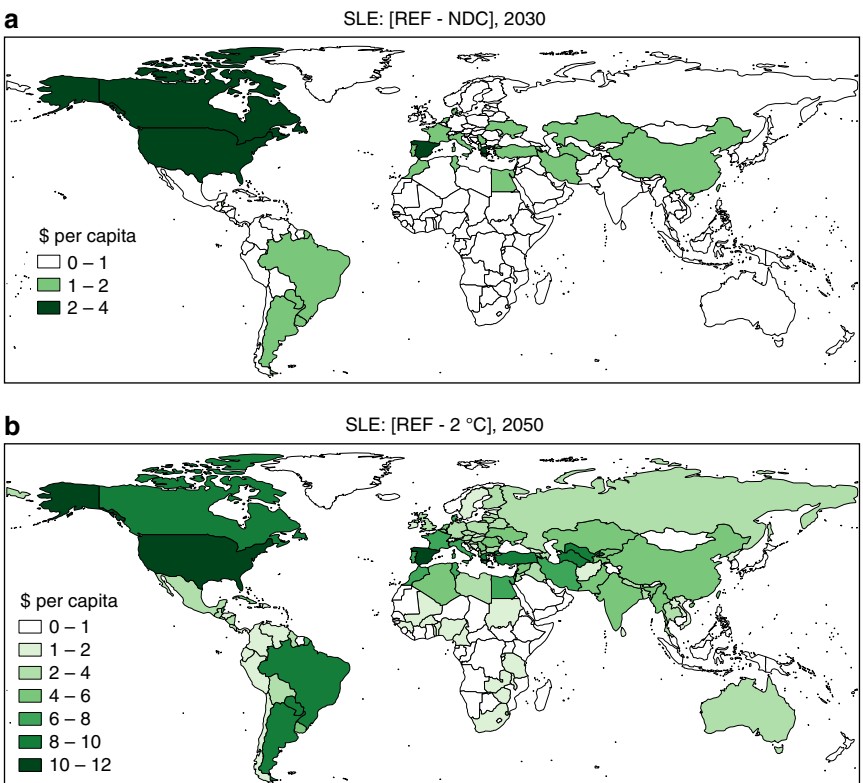

**Fig. 5** Ozone-related crop yield benefits due to climate policy in constant 2004–2006 dollars per capita. Results are shown for the case of Stringent Legislation (SLE) for air pollution. **a** Difference between the Reference (REF) and the NDC scenario in 2030. **b** Difference between the Reference and the 2 °C scenario in 2050. Valuation obtained by applying future crop productivity improvements compared to the Reference on the average gross production value of 2009–2013 and dividing by the population average of the same period

## Discussion

Climate policy's co-benefits related to air quality that are not covered here include the reduced cost of air pollution control measures[43–46], the avoided health-care expenditures[47–49], the nonmarket value of reduced morbidity, the implications of GHG reduction via land use for forest fires and related air pollution[50], indoor air quality and corresponding human health outcomes[51,52], the effects of acidification and eutrophication on ecosystems (SDG15), and the impact of air pollution on human capital formation and on-the-job performance[53,54]. Future work could shed light on these issues in the context of the Paris Agreement. The analysis presented here illustrates that the co-benefits of climate policy depend on the stringency of air pollution control measures and the valuation of avoided premature mortality, but additional sources of uncertainty can be found in each step of the methodology. Future research efforts could assess the combined uncertainty throughout the modelling chain based on a multi-model assessment. A better measurement of (exposure to) air pollution[55,56], the inclusion of additional health endpoints such as diabetes[57], and revised estimates of disease-burden[47,58] can further contribute to an improved understanding of the health impacts of air pollution.

The analysis presented here quantifies the local and global air quality co-benefits of climate policies (1) as pledged in the NDCs and (2) in a scenario where countries ratchet up ambition levels in order to curb global GHG emissions to reach the 2 °C temperature goal, complementing recent work on the health effects of a pathway compatible with 1.5 °C warming[59]. In addition to other co-benefits of climate action, such as improved human health through diet change[60] and enhanced ecosystem services via land use-based mitigation measures[61], the synergies with air quality call for an integrated policy perspective to enable progress in a broader context of sustainability, as represented by the Sustainable Development Goals. This paper takes climate policy objectives as a starting point, but long-term strategies to be submitted by 2020 in accordance with Article 4 (Paragraph 19) of the Paris Agreement can reach benefits beyond those presented here when an integrated policy is designed explicitly to balance trade-offs and ancillary benefits from the outset. Effective policymaking should therefore account for multiple externalities in the pricing of food and energy[62], consider the interplay of various policy design features, such as taxes and cap-and-trade mechanisms[63], and cover a broad range of pollutants, including short-lived climate forcers[64–68]. Future research should aim to show how policies that provide the right market incentives for technology trade-offs can exploit synergies and avoid lock-in effects in infrastructure by combining effective short-term air pollution control measures with an ambitious decarbonisation roadmap to maximise benefits for climate and human health simultaneously.

## Methods

**GHG emissions and energy system**. The energy and GHG emissions projections were done using the POLES-JRC model. POLES-JRC[69] is a global sectoral simulation model for the development of long-term energy demand and supply pathways with worldwide coverage. Projections are made on the basis of exogenous economic growth, demographic projections, and energy resources for each region, with prices driving the balance of supply and demand and international trade for each type of energy. POLES-JRC describes energy and emissions balances for 54 individual countries and 12 regions; it identifies 14 fuel supply branches and 15 energy demand sectors, and >40 energy technologies with endogenous technical progress. Energy-related GHG emissions are directly derived from the energy balances; emissions from the industry sector are calculated using marginal abatement curves derived from Environmental Protection Agency[70]; emissions for the agriculture and land use emissions are obtained by using data from the GLOBIOM model[71,72] and are sensitive to the carbon price. The scenarios presented here were developed using a methodology similar to that described in earlier work[27], with

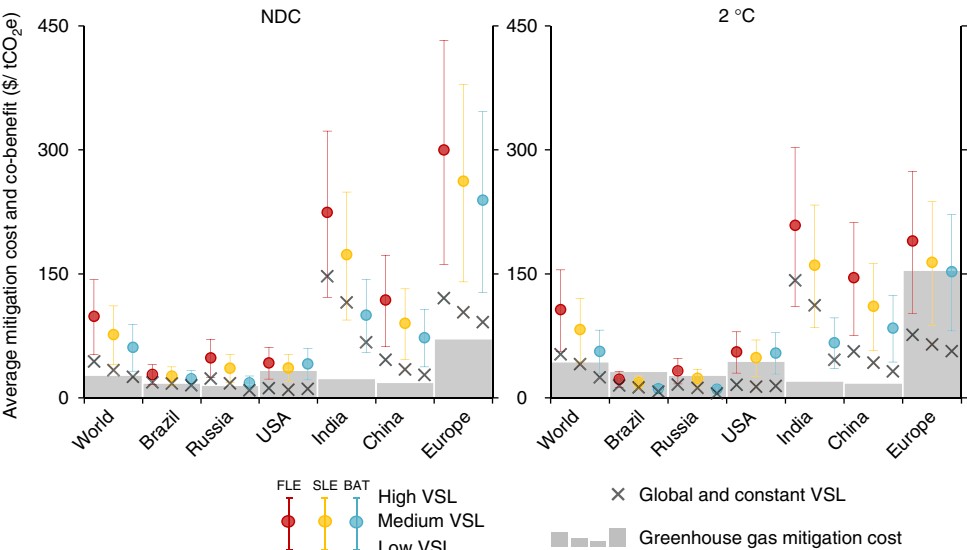

**Fig. 6** The value of the co-benefits of improved air quality due to climate polices in NDC and 2 °C scenarios. Values represent the average over 2015–2050 for Fixed Legislation (FLE), Stringent Legislation (SLE), and Best Available Technologies (BAT) air quality policies. Co-benefits include the value of avoided premature mortality as well as the co-benefits on the labour and agricultural markets via avoided work days lost and improved crop yields, respectively. The whiskers indicate high, medium, and low value of statistical life (VSL), heterogeneous across regions and time depending on GDP per capita. The black cross indicates results with the value of statistical life of 1.5 million US$(2005) constant across regions and over time. The shaded area presents the costs (change in welfare expressed as equivalent variation) of climate change mitigation policy over 2015–2050 and does not consider any co-benefits, nor does it include direct benefits of avoided impacts of climate change. Both cost and co-benefits are expressed per tonne of greenhouse gas emissions reduced excluding land use (change) and forestry

similar sources for the assumptions on macroeconomic development and the same set of policies considered for the REF, NDC, and 2 °C scenarios. The full detail on GHG emissions per region and scenario is presented in the additional results. The REF considers current and announced energy and climate policies only up to 2020. The NDC scenario assumes attainment of the objectives put forward in the NDC documents with the time horizon of 2025–2030. After 2030, we assume continued decarbonisation effort such that at the world (not regional) level the rate of decrease of the emissions intensity of the economy over 2030–2050 is the same to that of 2020–2030. Importantly, the NDC scenario has been developed by scanning all NDCs submitted to the UNFCCC and incorporating explicitly the policy details on renewables, biofuels, electric vehicles, etc. The 2 °C takes the NDC scenario as a starting point and lower limit and considers a 2011–2100 global carbon budget of 1100 GtCO$_2$. Regionally differentiated carbon prices reflect common but differentiated responsibilities as included in the United Nations Framework Convention on Climate Change, negotiated at the Rio Earth Summit in 1992 by assuming carbon price convergence only by the year 2050, with convergence speed depending on GDP per capita.

**Air pollutant emissions.** Air pollutant emissions, historical and projected, were obtained using emission factors data multiplied by activity data for each sectoral flow, endogenously produced by POLES-JRC. This was done for 35 sectoral flows (sectoral and fuel-specific energy demand, industrial activity, population) for each of the 6 pollutants (SO$_2$, NO$_x$, CO, BC, organic carbon, non-methane volatile organic compounds). The emission factors were derived from the GAINS model[73], using information from the ECLIPSE exercise[74].

**Temperature changes.** The corresponding ranges of temperature change, shown in Fig. 1 for all three climate scenarios, are derived from the probabilistic version of the MAGICC6 climate model[75] with Bayesian sampling of parameters from a prior probability distribution[76].

**Air pollution concentrations and mixing ratios.** To compute pollutant concentrations from pollutant emission scenarios, we use the TM5-FASST model[77], a reduced-form air quality assessment tool, built on pairwise emission–concentration sensitivities (so-called source–receptor coefficients (SRCs)) between 56 source regions and individual 1°×1° receptor grid cells for each relevant emitted pollutant or precursor.

The embedded sensitivities have been derived with the global two-way nested chemistry-transport model TM5[78] with year 2001 meteorology, from a large set of emission perturbation experiments, using Representative Concentration Pathway (RCP) year 2000 emissions as a reference[79], applying a −20% emission perturbation on each precursor and for each single source region. The TM5-FASST model delivers relevant exposure metrics (annual mean PM$_{2.5}$, highest occurring

6-monthly mean of daily maximum hourly ozone, crop-growing season mean daytime ozone). For the evaluation of arbitrary emission scenarios, the stored sensitivities are linearly extrapolated using an appropriate scaling factor, thus by-passing computationally expensive simulations and allowing for a fast screening of large sets of scenarios or scenario ensembles.

TM5-FASST takes as input annual anthropogenic emission strengths of individual pollutants and pollutant precursors (SO$_2$, NO$_x$, NH$_3$, primary PM$_{2.5}$ including BC and organic carbon, volatile organic components) aggregated at the level of 56 pre-defined source regions (as identified in Supplementary Figure 1) and produces approximate resulting pollutant grid maps of PM$_{2.5}$ and ozone, based on an implicit underlying spatial distribution of the emissions, originating in the gridded reference RCP emissions on which the SRCs are based. Examples of emission–pollutant source–receptor matrices include SO$_2$ to sulphate, NO$_x$ to nitrate and to ozone, NH$_3$ to ammonia, and BC to BC (as primary pollutant). The current study does not consider changes in natural PM$_{2.5}$ components (mineral dust, sea salt) to uniquely attribute changes in air quality to the climate policies. Ambient PM$_{2.5}$ is obtained as the sum of ammonium nitrate, ammonium sulphate, and primary emitted PM$_{2.5}$. Ozone is evaluated from NO$_x$ and volatile organic compounds, including the long-term feedback of methane on background ozone. In general, chemical processes leading to secondary pollutants involving multiple precursors are non-linear and the embedded linearisation in TM5-FASST may lead to biased estimates of PM$_{2.5}$ concentrations and ozone mixing ratios. However, validation studies[77] have shown that the linearity assumption in TM5-FASST holds sufficiently well for regionally averaged (population-weighted) PM$_{2.5}$ and ozone responses to precursor emission changes that deviate from −80% to +100% from the reference emissions.

As described in earlier work[77], TM5-FASST includes an optional urban increment adjustment for sub-grid PM$_{2.5}$ gradients. This adjustment was, however, not applied in the current analysis as it requires gridded sector-wise emission fields of the evaluated scenarios, which are not available in this study. Therefore, resulting PM$_{2.5}$ should be considered as lower bound values. For the health analysis, PM$_{2.5}$ concentrations from TM5-FASST have been adjusted to match the World Bank data (2010) on PM$_{2.5}$ exposure consistent with the Global Burden of Disease (GBD) 2015[80] and taking into consideration improved exposure estimation methods complemented by satellite and ground-level measurements[56].

As described above, TM5-FASST takes as input emissions aggregated at the level of the 56 pre-defined regions. For this study, native emission data were available at the aggregation level as listed in additional results, except for the European Union (28 countries) where data for individual countries were provided. The available aggregation was remapped to the 56 TM5-FASST regions by first downscaling POLES-JRC native regions to individual countries and subsequently re-aggregating countries to the 56 TM5-FASST source regions. The downscaling was done by individual IPCC sector, using RCP sector-segregated gridded emissions as a proxy to establish the weight coefficient of each country within a

larger region. Hence, for native region $R$ composed of countries $c_1$–$c_n$, the emission $E_{ps}(c_i)$ from country $c_i$ of precursor p in sector s is obtained as

$$E_{ps}(c_i) = f_{ps,i} \cdot E_{ps}(R) \tag{1}$$

where

$$f_{ps,i} = \frac{E_{ps}^{RCP}(c_i)}{\sum_{j=1}^{n} E_{ps}^{RCP}(c_j)} \tag{2}$$

and $E_{ps}^{RCP}(c_i)$ equals the RCP sector-specific emission summed over all grid cells of country $c_i$ and $\sum f_{ps,i} = 1$.

For each climate policy scenario in the current study, the closest matching RCP scenario was selected to guide the mapping of the regional emissions to TM5-FASST source regions. For the REF, NDC, and 2°scenario, we used RCP6.0, RCP4.5, and RCP2.6, respectively.

POLES-JRC sectors are mapped to the IPCC sectors as shown in Supplementary Table 45. In order to reduce artefacts associated with the downscaling and remapping of emissions between POLES-JRC and FASST regions, impacts are aggregated to the regional scale. Population-weighted regional pollutant exposure is obtained by interpolating TM5-FASST 1°×1° pollution grid maps to 0.5°×0.5° and overlaying them with SSP gridded population maps at the same resolution for the corresponding scenario year[81]. The regional population-weighted exposure to pollutant $P$ is then obtained as

$$\overline{P_{pop}} = \frac{\sum_i pop_i \cdot P_i}{\sum_i pop_i} \tag{3}$$

where index i runs over all grid cells of the considered region and $pop_i$ and $P_i$ are the population per grid cell and the grid cell pollutant concentration, respectively.

**Co-benefits on avoided premature mortality.** Here we use an impact pathway analysis or alternatively labelled a systems approach that quantifies the pollutant emissions from the moment they are released into the environment, followed by atmospheric dispersion, and removal by deposition and chemical transformation, and, finally, the impact on human health and the corresponding valuation. Five causes of premature death linked to the $PM_{2.5}$ pollution have been considered in this study: ischaemic heart disease, cerebrovascular disease (stroke), chronic obstructive pulmonary disease, lung cancer, and lower respiratory infection. For the health impact of ground-level ozone, we consider chronic obstructive pulmonary disease.

Health effect of changes in air pollution are calculated using epidemiological associations (relative risks (RRs)) linking ambient air concentration to specific health hazards in the general population. RR is defined as the ratio of health events in a risk group exposed to air pollution compared to a control group that is unexposed. RR of unity implies no risk difference between the two subpopulations.

ERFs are based on the GBD 2015[80]. These associations are distinguished by specific cause of death and in the case of cardiovascular mortality according to different age groups[37]. The $PM_{2.5}$ risk functions are non-linear with greater slope at lower ambient air exposures above the minimum threshold of 2.4 µg m⁻³, gradually levelling off at higher concentration values. The steepness varies according to illness and age group. For ozone mortality, the ERFs are shaped like a hockey stick, with no increase in the mortality risk up to a minimum concentration of 37.6 ppb (75 µg m⁻³), also known as the theoretically minimum risk exposure level, and thereafter the risk varies as a log-linear function of the concentration level (measured in ppb units) above the threshold. For pollutant p, cause of death c, country i and year t, the ERF translates concentration ($C$) to an RR factor:

$$RR_{p,c,i,t} = ERF_{p,c,a}(C_{p,i,t}) \tag{4}$$

This RR factor is the basis to calculate the Population Attributable Fraction (PAF), which measures the attributable share of the total burden of disease that is related to ambient air pollution:

$$PAF_{p,c,i,t} = 1 - \frac{1}{RR_{p,c,i,t}} \tag{5}$$

Note that the formula above is equivalent to the WHO definition (http://www.who.int/healthinfo/global_burden_disease/metrics_paf/en/) since we calculate premature mortality based on population-weighted country-level aggregate levels of concentration with the RR under ideal exposure to air pollution equal to unity. The number of premature deaths (PD) is then obtained by multiplying the PAF with the cause-specific Baseline Mortality (BM):

$$PD_{p,c,i,t} = BM_{c,i,t} * PAF_{p,c,i,t} \tag{6}$$

Although we interpret these deaths as linked to air pollution, we should note that air pollution acts in combination with other underlying diseases in the

population that ultimately result in premature mortality or otherwise loss of expected remaining life. Deaths solely caused by air pollution, i.e. initiated by air pollution in isolation of other health risk factors, are likely to be lower. In the above expression, the cause-specific BM is specified based on GBD 2015 as

$$BM_{c,i,t} = ABM_{i,t} * \frac{BM_{c,i,t=2015}}{\sum_c BM_{c,i,t=2015}} \tag{7}$$

in which the All-cause, natural Baseline Mortality (ABM) is obtained by multiplying UN forecasts of population (medium fertility scenario) with UN crude mortality rate projections[82] and a scaling factor to reconcile UN data with GBD 2015:

$$\delta_i = \frac{(All - cause\ deaths\ GBD)_{i,t=2015}}{(All - cause\ deaths\ UNWPP)_{i,t=2015}} \tag{8}$$

The numbers are presented by region in Supplementary Table 46.

**Co-benefits on lost work days.** Pollution-related illness was calculated assuming that the percentage change in the number of cases of morbidity between scenarios were proportional to changes in the projected mortality. In particular, lost work days from air pollution-related illnesses were calculated using a morbidity-to-mortality multiplier of 547 avoided lost work days per avoided premature mortality, derived from the WHO-HRAPIE[83] recommendations, based on earlier work[84], and applied in the context of EU Clean Air Package[85]. More specifically, based on the supporting document[86], we calculate a multiplier on the EU aggregate level:

$$\frac{Lost\ work\ days_{2010}}{Premature\ mortality_{2010}} = \frac{121378612\ days}{380196\ deaths} = 319\ lost\ work\ days\ per\ premature\ mortality \tag{9}$$

Next, this number is corrected by a calibration factor that reflects the use of GBD 2015 ERFs in this study relative to WHO-HRAPIE:

$$\frac{Premature\ mortality_{2015}^{HRAPIE\ ERF}}{Premature\ mortality_{2015}^{GBD\ ERF}} = \frac{441326\ deaths}{257544\ deaths} = 1.71 \tag{10}$$

The resulting morbidity-to-mortality multiplier is in the same order of magnitude as the multiplier used in other studies[87,88] of approximately 445 lost work days per premature mortality. One advantage of this approach is that the non-linear health response to air pollution is automatically accounted for. However, applying the same morbidity-to-mortality multiplier across space and time is a rough approximation that should be interpreted with caution and could be improved in future work by expanding the evidence base on air pollution-related morbidity and labour supply responses, as illustrated by recent work[89].

**Co-benefits on crop yields.** Translating changes in ground-level ozone mixing ratios to yield impacts is done through ERFs. These crop-specific functions relate ozone exposure to yield and are based on literature[90] for wheat, maize, rice, and soy. Three generic classes of ERFs were estimated for high, medium, and low sensitivity crops. Based on a meta-analysis[91], another 23 crop categories were allocated to these generic categories (see Supplementary Table 47). Using the changes in ozone mixing ratios across climate scenarios (keeping air pollution control scenario fixed to FLE, SLE, or BAT) and the ERFs, we obtain the yield impact in percentage terms by crop, climate scenario, air pollution control scenario, country, and year. Next, we aggregate crop productivity impacts across crops and regions using 5-year average (2009–2013) gross production values from FAOSTAT (http://www.fao.org/faostat/en/#data/QV).

**Integrated economic framework.** For the economic valuation of co-benefits, we follow a hybrid approach that combines market and nonmarket benefits, mirroring the methodology recently applied in the literature assessing the economic impacts of climate change[92]. The market co-benefits for labour markets through a reduction of lost work days due to illness ($PM_{2.5}$) and for agriculture markets via improved crop yields ($O_3$) are fed into the global economy-wide computable general equilibrium (CGE) model JRC-GEM-E3 to assess the broader economic impacts. Avoided lost work days translate into an expansion of the labour supply, while crop yield benefits are implemented through raised total factor productivity in the agricultural sector. The methodology builds on earlier work on air pollution[93] and allows inclusion of feedback effects via firms' supply chains, via households' income, and via international trade. The impact of changes in competitiveness on international trade may be especially important for agricultural crops that are traded on global markets, while general equilibrium effects could be substantial for the impact through changes in labour supply because a rise in income generates additional demand for goods. The valuation of avoided premature mortality does not enter the CGE model but is monetised by using the VSLs discussed in the next section. For mortality, earlier work[92] shows that the nonmarket component by far exceeds the market component (e.g. lost earnings).

The JRC-GEM-E3 model describes consumer and producer behaviour; represents government policies such as taxes, subsidies, transfers, and emission caps; captures endogenously the international trade flows based on (changes in) relative prices; and includes macro feedback mechanisms via forward and backward supply chain linkages and via labour market, wages, and employment effects. The model is designed to also estimate the cost of climate change mitigation policies, as presented in Fig. 6 and in earlier work[27]. In addition to common baseline pathways for economic growth, population projections, and GHG emissions, the POLES-JRC and JRC-GEM-E3 are fully harmonised in terms of GHG emissions and the regional electricity generation technology mix in the scenarios studied. A more extensive description of the model and the mathematical expressions can be found in the model documentation[94]. The regional aggregation used in this paper is shown in Supplementary Table 48. The metric best used to assess cost and benefits in this framework is welfare changes in terms of equivalent variation, which measures the difference in expenditure needed to bridge the change in utility levels at base prices. The costs and co-benefits presented in this paper are undiscounted.

**VSL across regions and time**. The VSL, taking into account heterogeneity across regions i and over time t, is specified in relation to real income per capita ($I_i^t$) in the following way:

$$VSL_i^t = VSL_{USA}^{2005} * \left( \frac{I_i^t}{I_{USA}^{2005}} \right)^{0.8} \tag{11}$$

In this equation, $I_i^t$ is expressed as GDP per capita at purchasing power parity. The income elasticity is chosen to be 0.8, which is more conservative than some earlier studies[10] but still within the range recommended by the OECD[95]. We should note here that some studies[96] discuss income elasticity values >1, which would lower the valuation of avoided premature mortality for regions with GDP per capita lower than the level in the USA in 2005. The low and high values for $VSL_{USA}^{2005}$ are based on values found in previous literature[10], while the medium value provides an intermediate case. The resulting values of statistical life are given in Supplementary Table 43. The alternative case with a VSL of 1.5 million US$(2005) that is constant over time is included to address ethical concerns, for comparability across regions, and because from the perspective of a global policymaker, introducing distributional weights that reflect inequality preferences in welfare aggregation would counteract heterogeneity in VSL across regions[97].

**Disclaimer**. The views expressed are purely those of the authors and may not in any circumstances be regarded as stating an official position of the European Commission.

## Data availability

The data that support the findings of this study are available from the corresponding author upon reasonable request.

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

## Author contributions

K.K., R.V.D., J.V.S., and T.V. did the energy, atmospheric, health, and economic modelling, respectively. R.V.D. and T.V. calculated the crop yield impacts. T.V. wrote the paper and created the figures. T.V., A.K., and B.S. conceived the study. M.H. and all other authors contributed to the text.

## Additional information

**Competing interests:** The authors declare no competing interests.

