## [Peer Review File · Nature Communications]

Reviewers' comments:

Reviewer #1 (Remarks to the Author):

The paper revisits the issue of air quality co-benefits of climate action, while carrying out an analysis of such co-benefits in context of the Nationally Determined Contributions (NDCs) under the Paris agreement and further action to reach the 2°C target. The paper uses the 'impact pathway approach' and provides estimates of the impact of NDCs and further actions on the changes in 1) emissions of air pollutants, 2) energy use (by energy source), 3) surface level concentration of PM2.5 and ozone, 4) premature mortality and work-day loss; 5) agricultural crop yields, and 6) the total monetized benefit (i.e. the value of the co-benefits included). Results are given for the World and for a range of countries/world regions. Estimates of air pollution emission reductions in the scenarios are taken from previous literature (IIASA, 2015). For most scenarios (that assume different ambitions in the baseline level of air pollution control), co-benefits are found to exceed the costs of climate change mitigation policies. In this respect the paper confirms a range of previous papers. Despite a lack of substantial novelty, the paper is a useful and topical contribution as it explicitly addresses the Paris agreement. Overall, the paper is well-written and concise, and is easy to follow and figures are generally clear and well designed.

I do, however, have some concerns related to the economic analysis, described in the following:

1. The description of how the calculation of Lost work days (LWD) was carried out is very brief and not convincing. It refers to WHO (2013) and Ostro (1978), but these only indicate the Exposure-response function for LWD. I cannot see how the "morbidity-to-mortality" multiplier of 547 is derived from these sources.
2. The authors use an economy-wide computable general equilibrium model (CGE) to estimate the economic benefit of avoided LWD and avoided crop loss. This seems ok, except that, as mentioned above, I cannot see the justification for the estimation of the number of LWD. My main concern is that mortality benefits are estimated using the Value of Statistical Life approach. The authors need to explain the hybrid approach to economic assessment (CGE modelling and VSL), and how they justify adding the mortality benefit (based on VSL) and benefits from avoided LWD and crop loss (based on changes in equivalent variation as estimated by the CGE).

Minor comments/suggestions are:

- Figure 2 lists IPCC AR5 WGIII (IIASA, 2015) as the source of the data, but this reference is not provided. The title of the y-axis is only showed for the last figure and not placed in a way that indicates it applies to all.
- First line, page 4, is confusing. Policies do not derive from underlying drivers (affect the same drivers.. better?)
- Line 3 page 4: I suggest to use the word 'change' instead of 'move'
- Figure 3: in the last figure the country borders below the violet field do not show (eg India), perhaps dilute color.
- Line 5-6 page 5: Lung cancer is a separate health end-point, hence better to say '...mortality due to cardiovascular and respiratory diseases and lung cancer.
- Figure 4: Suggest to denote the y-axis of the larger figure 'Avoided deaths (1000)' or similar (not people). The last sentence in the caption is confusing, what is the unit of the y-axis on the smaller figures, still Avoided deaths (1000)? The GBD (2015)reference is not given.
- Figure 5: Related to my comments above regarding the economic estimates, were the benefits of avoided LWD and crop loss subtracted from the welfare cost of climate change mitigation policy over 2015-2050? This should be described in the text describing this figure.
- 4th line from below, page 6: unclear sentence, do you mean ..a reduction if pollution in the agricultural and transport sectors?
- First paragraph, page 7: In the discussion of what is not included in the co-benefit assessment, you could also mention potential additional benefits related to avoided household air pollution from solid household fuel. These benefits may be large as the intake fraction of emissions is high.

Suppl. Mat:

- Page 17, the subscripts in the equations are not defined
- Page 18: Please provide link/reference to FAOSTAT data.
- Page 18, 8th line from below: Marked – not market
- Page 20: You may add an elasticity estimate greater than 1.0. Elasticities >1.0 are suggested by research on the relationship between long-term economic growth and the VSL, by cross-country comparisons, and by new research that estimates the VSL by income quartile. Studies also suggest that the elasticity varies by income level; i.e. lower income levels are associated with higher income elasticities of VSL (see Hammitt and Robinson 2011), who report elasticities in the range 1.5–3.0 in longitudinal studies.
- Table p 26 and onwards: Share in total primate energy $_(\%)_$

Reviewer #2 (Remarks to the Author):

This paper propagates the National Determined Contributions (NDCs) in the Paris agreement to their impact on Air Quality improvement. It also explores other Climate Change scenarios and Air Quality story lines. The paper radiates the strong message that the financial co-benefits of climate change policies clearly compensate the costs. In that sense, the paper clearly targets a broad audience and the current journal is therefore suitable.

Although parts of the methods have been applied elsewhere, the current application is, as far as I can tell, new and original.

However, the language used in the paper is quite technical, and assumes deep insight in the often-complicated implications of AQ and Climate policies. Also, some more words of caution are needed about the huge uncertainties (even unknown unknowns) associated with the scenario work. Terms like VSL should be better explained in the main text, and in general the link between text and figures could be improved.

In conclusion, if the authors manage to smooth the language and to explain the main points I mention below, I see no major obstacles in publication.

Some main points:

Naming and consequences of the Air Quality Storylines are somewhat unclear. "The Fixed Legislation (FLE) scenario considers no additional implementation of air pollution abatement". This is the less stringent scenario concerning air pollution abatement. The authors therefore claim that this scenario is: "providing a benchmark case to determine the maximum extent of possible co-benefits.". What I understand from this is that Climate Policy will have a larger co-benefit for air pollution abatement. Reversely, little further AQ improvement from Climate Policy is expected in the Best Available Technology scenario (BAT). In that sense, I understand the symbols in figure 2: "Symbols represent the results for the Stringent Legislation air quality scenario, while vertical lines indicate the results for the Fixed Legislation (high) and Best Available Technologies (low) air quality scenarios", since more Climate Policy causes less AQ benefits when clean air technology is already in place.

When I now try to understand figure 1, I notice that the BAT scenario has higher projected T-changes compared to SLE, and FLE lower projected T-changes. I think some more explanation is required here. Maybe this is trivial for an audience that works with these scenarios all day, but the targeted audience is broader here. There is also a coupling between AQ policy on Climate Policy, and I think this complicates matters. For instance, with stringent AQ policy (BAT), climate cooling

by aerosols will diminish. Is this what I see in Figure 1? I think most of these issues can be remedied by small clarifications in the text, and some editing by non-specialists in the field.

In the discussion of figure 5, the main text should also explain the use of VLS, since this critically influences the results. Also, in the grey "costs" bar, the authors do not account for the extra costs for BAT, compared to FLE, which I think would be a fair way of presenting the results.

Minor points:

Page 4, near bottom: Figure3 misses a space.

Page 4, one line lower: after "chemistry reactions of pollutants", some explanations about the (simplified) methodology would be in place (or a reference to the Appendix).

Page 5: "globally in the year 2030 compared to current climate policies". Confusing use of 2030 and 2050. This makes the numbers difficult to compare.

Caption figure 4: " ..(in thousands) for six regions, with the vertical axes ranging from 0 to 0.4 x 10⁻³ x population of the region in 2050.". Unclear. I see all vertical axis being different, so what does this mean?

ppbV: unit is obsolete. Unit "ppb" is preferred.

See: https://en.wikipedia.org/wiki/Parts-per_notation

Related: Units ppb express a mixing ratio, not a concentration. So, the text and supplement should mention mixing ratio instead of concentration.

Reviewer #3 (Remarks to the Author):

In this study, the authors quantified the co-benefits on air quality, premature mortality, crop yields as well as labor loss from the climate policies indicated under the Paris Agreement. Though the concept of the co-benefits are not very unique, as also discussed by the authors, the focus targeting the Paris Agreement will intrigue some interests among the community. However, the manuscript was not well structured, and did not make clear for the discussion and explanation of the results. Specifically, the manuscript is also vague on the methodology. How the connection between air quality improvement and crop yields as well as the labor loss are not well discussed in the methodology. How did the authors project the changes of baseline mortality rates and population in the future? There are Table (2.1) in the supporting material, but the authors did not explain their methods.

In the abstract, the authors summarized the avoided premature deaths from different climate policies (NCDs & under 2°C) both short-term and long-term. Despite the conclusion is poorly summarized, it does not make a very good comparison either: first, these two climate policies are compared at different decades (2030 & 2050s); second, how the ranges of the premature deaths (0.7-1.5 million) are derived are unclear. From the manuscripts, it seems to me that the range come from the different air quality policies. Moreover, it will be beneficial if the authors can report the 95% for all their estimation. The manuscript also incorporated some analysis on the labor loss which were not discussed in the abstract in justifying the co-benefits.

In page 3, the authors discussed that incorporating different air quality policies could broaden the aspects of this study, by "This hypothetical benchmark identifies how structural changes induced by climate policy can improve air quality beyond what is possible by air pollution abatement

technologies alone". However, the authors failed to discuss the results from this perspective. The reduced premature mortality, increased crop yields as well as the labor loss under the co-benefits of climate policies (NDCs-REF or 2°C-REF at 2030&2050s) should be compared among these three different air quality policies. Otherwise, the authors could not make this claim, and then make their objectives vague.

The manuscript missed the main discussion on co-benefits. All through the figures in the paper as well as the supporting, I do not see straight-forward discussions on the co-benefits.

Specific comments:

Page 1: West et al., (2013, Nature Climate Change) has a more recent estimation on the monetized co-benefits. Consider to update.

Page 2: Change "The 2°C scenario considers policies that that is consistent with " to "that that are consistent with".

Figure 1 in the manuscript is very like the authors' another paper published in 2016: (Vandyck et al., 2016, <https://doi.org/10.1016/j.gloenvcha.2016.08.006>). Please consider make reference or redo the figure.

Page 3 Fig. 2: make it consistent when presenting the results for the air pollutants: Black carbon – BC; Carbon monoxide—CO.

Page 4: "On a regional level, air quality co-benefits generally" I didn't see how the authors derive this conclusion. It is not obvious to me.

Page 4 Fig. 3: when defining the co-benefits, it is better to compare 2°C/NDCs-REF, instead of RFE-2°C/NDCs.

In supporting, Page 17: the Population Attributable Fraction (PAF) was not defined in an appropriate way. See the definition from WHO website:

http://www.who.int/healthinfo/global_burden_disease/metrics_paf/en/

The paper could also benefit if the authors could add some spatial analysis for the co-benefits or air quality analysis.

Reply to reviewer #1

Major comments

1. "The description of how the calculation of lost work days (LWD) was carried out":

We agree that the methodology description was too vague and have improved the text in Supplementary Information correspondingly. The morbidity-to-mortality multiplier is calculated based on previous work, including a correction for the use of non-linear exposure-response functions in this study. The revised text is more explicit and transparent on the calculation, discusses benefits and drawbacks, refers to other studies that use this approach, and points out the need for further research:

"Pollution-related illness was calculated assuming that the percentage change in the number of cases of morbidity between scenarios were proportional to changes in the projected mortality. In particular, lost work days from air pollution-related illnesses were calculated using a "morbidity-to-mortality multiplier" of 547 avoided lost work days per avoided premature mortality, derived from the WHO-HRAPIE (82) recommendations, based on earlier work (83), and applied in the context of EU Clean Air Package (84). More specifically, based on the supporting document (85), we calculate a multiplier on the EU aggregate level:

$$\frac{\text{Lost work days}_{2010}}{\text{Premature mortality}_{2010}} = \frac{121378612 \text{ days}}{380196 \text{ deaths}} = 319 \text{ lost work days per premature mortality.}$$

Next, this number is corrected by a calibration factor that reflects the use of GBD2015 ERFs in this study relative to WHO-HRAPIE:

$$\frac{\text{Premature mortality}_{2015}^{\text{HRAPIE ERF}}}{\text{Premature mortality}_{2015}^{\text{GBD ERF}}} = \frac{441326 \text{ deaths}}{257544 \text{ deaths}} = 1.71.$$

The resulting morbidity-to-mortality multiplier is in the same order of magnitude as the multiplier used in other studies (86-87) of approximately 445 lost work days per premature mortality. One advantage of this approach is that the non-linear health response to air pollution is automatically accounted for. However, applying the same morbidity-to-mortality multiplier across space and time is a rough approximation that should be interpreted with caution and could be improved in future work by expanding the evidence base on air pollution-related morbidity and labour supply responses, as illustrated by recent work (88)."

2. "Explain the hybrid approach to economic assessment (CGE modelling and VSL)":

Along with additional text to clarify the use of VSL (as also requested by other reviewers), we have added a better explanation of the approach and a better connection to the existing literature (Hsiang et al., 2017, *Science*):

"Whereas crop yields and work days are reflected by a market value, the avoided premature mortality is largely a nonmarket co-benefit, evaluated here by using the Value of Statistical Life (VSL). The low, medium, and high VSLs are heterogeneous across regions and grow with income over time (more details in Supplementary Information), as empirical studies typically find that the willingness-to-pay for health risk reductions varies with income. For comparison, we also include the results with a VSL that is homogeneous across regions and fixed in time at a more conservative value."

"For the economic valuation of co-benefits, we follow a hybrid approach that combines market and nonmarket benefits, mirroring the methodology recently applied in the literature assessing the economic impacts of climate change (60). The 'market' co-benefits for labour markets through a reduction of lost work days due to illness (PM_{2.5}) and for agriculture markets via improved crop yields (O₃) are fed into the global economy-wide computable general equilibrium (CGE) model JRC-GEM-E3 to assess the broader economic impacts. Avoided lost work days translate to an expansion of the labour supply, while crop yield benefits are implemented through raised total factor productivity in the agricultural sector. The methodology builds on earlier work on air pollution (49) and allows inclusion of feedback effects via firms' supply chains, via households' income, and via international trade. The impact of changes in competitiveness on international trade may be especially important for agricultural crops that are traded on global markets, while general equilibrium effects could be substantial

for the impact through changes in labour supply because a rise in income generates additional demand for goods. The valuation of avoided premature mortality does not enter the CGE model but is monetised by using the Values of Statistical Life discussed in the next section. For mortality, earlier work (60) shows that the nonmarket component by far exceeds the market component (e.g. lost earnings)."

Furthermore, we list "the nonmarket value of reduced morbidity" in the caveats paragraph.

Minor comments have been addressed by the following edits:

- All axes in Figure 2 are now labelled to make the figure clearer, and the reference to IIASA (2015) has been added in the reference list.

- Confusing text on "underlying drivers" has been improved

- Line 3 page 4 (in first submission) has been adjusted

- Lung cancer is now mentioned as a separate health end-point

- Figure 4: y-axis labels have been changed following the suggestion, the caption text has been improved, and the reference to GBD (2015) has been added.

- Figure 5: caption has been improved, explicitly stating that the mitigation cost excludes (co-)benefits

- 4th line from below, page 6 (in first submission) has been modified

- Indoor air pollution from solid household fuel is now included in the caveats, with references to

Smith, K. R., Bruce, N., Balakrishnan, K., Adair-Rohani, H., Balmes, J., Chafe, Z. et al. (2014). Millions dead: how do we know and what does it mean? Methods used in the comparative risk assessment of household air pollution. *Annual review of public health*, 35, 185-206.

Goldemberg, J., Martinez-Gomez, J., Sagar, A., & Smith, K. R. (2018). Household air pollution, health, and climate change: cleaning the air. *Environmental Research Letters*, 13(3), 030201.

The **Supplementary Information** has been improved by the following edits:

- Subscripts in the equations on page 17 (in first submission) are defined as pollutant p , cause of death c , country i and year t

- Adding a link to the FAOSTAT data

- More explanation to clarify what we mean with 'market' co-benefits

- We have added a comment about income elasticities greater than 1 and refer to the work of Hammitt and Robinson (2011). We believe that by exploring 4 different options for the Value of Statistical Life (low, medium, high, and constant across time and space) we answer to these authors' call to "avoid relying on a single VSL". The following text has been added:

"We should note here that some studies (54) discuss income elasticity values higher than one, which would lower the valuation of avoided premature mortality for regions with GDP per capita lower than the level in the USA in 2005."

- Adding (%) where relevant in the table headings

Reply to reviewer #2

Major comments

1. "Naming and consequences of the Air Quality Storylines are somewhat unclear"

We have added clarifications in the scenario description. In addition, we now discuss the implications for the results in more detail, as also requested by Reviewer #3.

"The *Fixed Legislation (FLE)* scenario considers no additional implementation of air pollution abatement technologies from 2010 onwards. In combination with a *Reference* climate policy, this assumption implies that economic and population growth lead to increasing global emissions over time for all air pollutants, with the exception of carbon monoxide, for which a rising trend is offset by ongoing progress in energy technology and corresponding efficiency improvements. Because high levels of air pollution provide a broad base for reductions, the estimates of the co-benefits of climate policy derived under the assumption of *Fixed* air quality *Legislation* will be considered here as an upper bound."

"In the *Best Available Technologies (BAT)* scenario, countries fully adopt the maximum technically feasible air pollutant emission reduction technologies by 2030. This hypothetical benchmark identifies how structural changes induced by climate policy can improve air quality beyond what can be expected by end-of-pipe air pollution abatement technologies alone, and will be used here to quantify a lower bound for the co-benefits."

"For organic carbon and carbon monoxide in particular, Figure 2 shows that not all models in the IPCC's Fifth Assessment Report (29) agree on the sign of the change in emissions on the global level, indicating uncertainty in the estimates. Figure 2 furthermore indicates the sensitivity of co-reductions with respect to the implemented air pollution control technologies. Low air pollutant emission intensities (*BAT*) in the benchmark in key climate change mitigation sectors reduce the scope for co-benefits, as can be seen from the sulphur dioxide emission reductions in India, for instance, where the implementation of *BAT* implies less polluting coal-fired electricity generation facilities."

2. "BAT scenario has higher projected T-changes compared to SLE, and FLE lower projected T-changes. I think some more explanation is required here"

We agree and have added the following explanation in a new paragraph, with references to the corresponding literature:

"Air pollutants affect global mean temperature changes, as well as regional variations around the average and precipitation patterns (Arneth et al., 2009; Samset, 2018). The effect of air pollution control on climate change is a priori unclear, because the radiative forcing of some pollutants, such as black carbon, is positive, while other pollutants (NO_x, SO₂) have a cooling effect on the climate. In addition to the median estimate of global average temperature change and the 50% probability bounds for the *SLE* scenario, Figure 1 (dotted lines) includes the median estimate in case of less (*FLE*) and more (*BAT*) stringent air pollution control measures. The figure shows that the temperature change relative to pre-industrial levels in the *Reference* climate policy under more ambitious air pollution controls (*BAT*) exceeds the central case estimate (*SLE*) by 0.08°C in 2100, while the higher end of the air pollution projection (*FLE*) implies temperature changes that are 0.32°C below the *SLE* case in 2100. Hence, the end-of-pipe reduction of air pollutants with a cooling effect outweighs the decrease in air pollutants that contribute to global warming in our scenarios, leading to a net upward effect on global mean temperatures, in line with other work (Rogelj et al., 2014; Hienola et al., 2018). This result does not consider the effect of air pollution controls on greenhouse gas emissions (which do not vary across air pollution control assumptions here) and is less pronounced in more ambitious climate mitigation scenarios (because co-emitted air pollutants are also reduced). Compared to the *Reference*, the decrease in greenhouse gas emissions in the 2°C scenario implies larger reductions in global mean temperature in the case of cleaner air (1.84°C under *BAT* vs.

1.62°C under *FLE*, *REF* – 2°C in 2100), since co-reduction of cooling aerosols plays a smaller role under stringent air pollution control."

3. "explain the use of VSL"

This is indeed a key element and deserves more attention. We have added clarification in the main text on the use of the VSL, and more background on the economic assessment in the Supplementary Information:

"Whereas crop yields and work days are reflected by a market value, the avoided premature mortality is largely a nonmarket co-benefit, evaluated here by using the Value of Statistical Life (VSL). The low, medium, and high VSLs are heterogeneous across regions and grow with income over time (more details in Supplementary Information), as empirical studies typically find that the willingness-to-pay for health risk reductions varies with income. For comparison, we also include the results with a VSL that is homogeneous across regions and fixed in time at a more conservative value. Compared to an approach based on years of life lost, the use of the same VSL for all deaths may bias the valuation upwards when the population affected is characterised by relatively old age, poor health conditions, and comparably short life expectancy."

"For the economic valuation of co-benefits, we follow a hybrid approach that combines market and nonmarket benefits, mirroring the methodology recently applied in the literature assessing the economic impacts of climate change (60). The 'market' co-benefits for labour markets through a reduction of lost work days due to illness (PM_{2.5}) and for agriculture markets via improved crop yields (O₃) are fed into the global economy-wide computable general equilibrium (CGE) model JRC-GEM-E3 to assess the broader economic impacts. Avoided lost work days translate to an expansion of the labour supply, while crop yield benefits are implemented through raised total factor productivity in the agricultural sector. The methodology builds on earlier work on air pollution (49) and allows inclusion of feedback effects via firms' supply chains, via households' income, and via international trade. The impact of changes in competitiveness on international trade may be especially important for agricultural crops that are traded on global markets, while general equilibrium effects could be substantial for the impact through changes in labour supply because a rise in income generates additional demand for goods. The valuation of avoided premature mortality does not enter the CGE model but is monetised by using the Values of Statistical Life discussed in the next section. For mortality, earlier work (60) shows that the nonmarket component by far exceeds the market component (e.g. lost earnings)."

4. "the authors do not account for the extra costs for BAT, compared to FLE, which I think would be a fair way of presenting the results "

It is true that we did not account for the (avoided) costs of air pollution control, which is now stated explicitly in the text. We are studying here the costs and co-benefits of climate policy, and therefore believe that the comparison between both is consistent. We did not study the costs and (direct) benefits of air pollution policy, and believe this is beyond the scope of this paper (a global cost-benefit assessment of air quality policies is a challenging task, which requires information on the costs and abatement potentials of end-of-pipe air pollutant controls). For a given assumption on air pollution control policy, we compare across climate policy scenarios. However, it is true that targets for air quality may be reached at lower cost when climate policy is more ambitious. This caveat – the co-benefit of reduced air pollution control cost – is acknowledged, with more literature references in the revised version of the paper. Furthermore, we have added the grey cost bars in the figure legend under the title "GHG Mitigation cost", in order to avoid confusion.

"In order to attribute the impacts uniquely to climate policy, we derive the costs (as well as the co-benefits) of climate policy by comparing the *NDC* and 2°C scenario with the *Reference* under the same assumption for air pollution control (e.g. *REF-FLE* vs. *NDC-FLE*) and therefore exclude the costs of air pollution control technologies."

"Climate policy's co-benefits related to air quality that are not covered here include the reduced cost of air pollution control measures (34-37)"

"Using the changes in ozone mixing ratios across climate scenarios (keeping air pollution control scenario fixed to *FLE*, *SLE*, or *BAT*) and the exposure-response functions, we obtain the yield impact in percentage terms by crop, climate scenario, air pollution control scenario, country, and year."

5. "words of caution are needed about the huge uncertainties (even unknown unknowns)"

Valid point. We now make an explicit statement about the uncertainty in the Discussion section. In addition, we highlight some of the more recent progress in the field.

"The analysis presented here illustrates that the co-benefits of climate policy depend on the stringency of air pollution control measures, but additional sources of uncertainty can be found in each step of the methodology. Future research efforts could assess the combined uncertainty throughout the modelling chain based on a multi-model assessment. A better measurement of (exposure to) air pollution (46-47), the inclusion of additional health endpoints such as diabetes (48), and revised estimates of disease-burden (39, 49) can further contribute to an improved understanding of the health impacts of air pollution."

Furthermore, we indicate the uncertainty in our analysis by comparing with other models in the community:

"For organic carbon and carbon monoxide in particular, Figure 2 shows that not all models in the IPCC's Fifth Assessment Report (29) agree on the sign of the change in emissions on the global level, indicating uncertainty in the estimates."

Minor comments

* Now more background on "chemistry reactions of pollutants" for a broad audience, plus a reference to the Supplementary Information:

"atmospheric chemistry reactions of pollutants (Supplementary Information). Hence, particulate matter includes both direct emissions from primary sources, such as black carbon and organic matter, and secondary PM that derives from emissions of NH₃, NO_x, SO₂, and VOCs (volatile organic compounds). Ozone is formed by the reaction of precursor gases NO_x, VOCs, and CO, in the presence of sunlight."

* "Confusing use of 2030 and 2050" corrected

* Caption of Figure 4 has been clarified

* All occurrences of ppbV have been replaced by ppb

* We now use the term "mixing ratio" for ozone

Reply to reviewer #3

Major comments

1. "manuscript was not well structured, and did not make clear for the discussion and explanation of the results"

We have added subtitles to clarify the structure. We believe the revised text does a better job in explaining the results by adding a section on air-climate interactions, by adding a new section and figure on the agricultural co-benefits, by making several clarifying adjustments in the text, and by expanding the Supplementary Information (in particular for work lost days and agriculture co-benefits). Furthermore, the discussion section of the paper has been improved.

2. "connection between air quality improvement and crop yields as well as the labor loss are not well discussed in the methodology"

We agree and have improved the manuscript correspondingly. The crop yield co-benefits now get more attention in the main text with a separate paragraph and figure dedicated to agriculture, and the Supplementary Information for both work lost days and crop yields is now more transparent on the calculations. New tables have been added in Supplementary Information sections 1.5.3 and 2.7.3.

3. "How did the authors project the changes of baseline mortality rates and population in the future?"

Projections of baseline mortality rates and population are based on United Nations forecasts, World Population Prospects: The 2015 Revision. This is mentioned in the Supplementary Information (Section 1.5.1), where we have added the corresponding reference. We had included the projections of population and baseline mortality rates in the Supplementary Information to enhance the transparency of our research.

4. "first, these two climate polices are compared at different decades (2030 & 2050s); second, how the ranges of the premature deaths (0.7-1.5 million) are derived are unclear"

The abstract and main text have been re-written based on these comments, now comparing *NDC* and 2°C in 2030, and clarifying that the range derives from different assumptions on air pollution controls. In the description of results, we maintain the focus on 2030 for the *NDC* scenario, which we believe is justified because this is the time horizon in most of the Nationally Determined Contributions submitted to the UNFCCC. The revised abstract now contains the following wording:

"As such, the NDCs could avoid between 71 and 99 thousand premature deaths annually in 2030 compared to a reference case, depending on the stringency of direct air pollution controls. Strengthening the ambition of climate change mitigation policies to limit global warming to below 2°C by the end of the century raises the number of avoided premature deaths from air pollution to 178-346 thousand annually in 2030, and up to 0.7-1.5 million in the year 2050."

"Because high levels of air pollution provide a broad base for reductions, the estimates of the co-benefits of climate policy derived under the assumption of *Fixed* air quality *Legislation* will be considered here as an upper bound. "

"This hypothetical benchmark [*BAT*] identifies how structural changes induced by climate policy can improve air quality beyond what can be expected by end-of-pipe air pollution abatement technologies alone, and will be used here to quantify a lower bound for the co-benefits."

5. "it will be beneficial if the authors can report the 95% for all their estimation"

We agree that there is substantial uncertainty underlying the analysis, starting from population and economic growth projections, and going up to the health impacts of air pollution. Our analysis addresses the uncertainty that stems from different air pollution control policies in the future and considers a range of assumptions for the valuation of avoided premature mortality. We believe these are two key aspects for discussing the sensitivity of the results. However, we believe that analysing the full cascade of uncertainties throughout the chain of models would be best addressed by a multi-model ensemble, an exercise that goes beyond the scope of this analysis. This is now acknowledged in the discussion section of the main text. Furthermore, we point out more areas for improvement, including references to recent literature.

"The analysis presented here illustrates that the co-benefits of climate policy depend on the stringency of air pollution control measures and the valuation of avoided premature mortality, but additional sources of uncertainty can be found in each step of the methodology. Future research efforts could assess the combined uncertainty throughout the modelling chain based on a multi-model assessment. A better measurement of (exposure to) air pollution (46-47), the inclusion of additional health endpoints such as diabetes (48), and revised estimates of disease-burden (38, 49) can further contribute to an improved understanding of the health impacts of air pollution."

In addition, including the results of other models in Figure 2, based on IPCC AR5, gives an indication of the uncertainty. This is now explicitly mentioned:

"For organic carbon and carbon monoxide in particular, Figure 2 shows that not all models in the IPCC's Fifth Assessment Report (28) agree on the sign of the change in emissions on the global level, indicating uncertainty in the estimates."

6. "co-benefits of climate policies (NDCs-REF or 2°C-REF at 2030&2050s) should be compared among these three different air quality policies" [*FLE*, *CLE*, and *BAT*]

The language explaining how we use the air pollution control scenarios was indeed unclear, and has been improved in the revised version. We are studying the co-benefits of climate policy, and therefore only compare across climate scenarios under the same air pollution control assumption, e.g. *NDC-FLE* versus *REF-FLE*, or *NDC-BAT* versus *REF-BAT*. This is now mentioned more clearly in various places in the text:

"The *Fixed Legislation (FLE)* scenario considers no additional implementation of air pollution abatement technologies from 2010 onwards. In combination with a *Reference* climate policy, this assumption implies that economic and population growth lead to increasing global emissions over time for all air pollutants, with the exception of carbon monoxide, for which a rising trend is offset by ongoing progress in energy technology and corresponding efficiency improvements. Because high levels of air pollution provide a broad base for reductions, the estimates of the co-benefits of climate policy derived under the assumption of *Fixed* air quality *Legislation* will be considered here as an upper bound. A gradual adoption and diffusion of air pollution control measures is included in the *Stringent Legislation (SLE)* scenario, which better reflects ambitious recent policy objectives in fast-growing countries such as China. In the *Best Available Technologies (BAT)* scenario, countries fully adopt the maximum technically feasible air pollutant emission reduction technologies by 2030. This hypothetical benchmark identifies how structural changes induced by climate policy can improve air quality beyond what can be expected by end-of-pipe air pollution abatement technologies alone, and will be used here to quantify a lower bound for the co-benefits."

"In order to attribute the impacts uniquely to climate policy, we derive the costs (as well as the co-benefits) of climate policy by comparing the *NDC* and 2°C scenario with the *Reference* under the same assumption for air

pollution control (e.g. *REF-FLE* vs. *NDC-FLE*) and therefore exclude the costs of air pollution control technologies."

"Using the changes in ozone mixing ratios across climate scenarios (keeping air pollution control scenario fixed to *FLE*, *SLE*, or *BAT*)"

7. "discussion on co-benefits" + "add some spatial analysis for the co-benefits"

We have added a section on the agricultural co-benefits, including a map that visualizes the distribution of the co-benefits across space. Furthermore, the Supplementary Information describing the methodology for work lost days and agriculture has been expanded, and additional results for the agricultural impacts by crop type have been added.

"Ground-level ozone penetrates leaves and hinders plant growth, thereby affecting agricultural productivity (38-39). By reducing ozone precursor emissions, global climate policy can improve crop yields. Using exposure-response functions for seven crop types (Supplementary Information), we calculate the crop yield impact of climate policy-induced reductions in tropospheric ozone mixing ratios. Projecting future changes in crop yield onto current values of agricultural output, we obtain the impacts shown in Figure 5, where monetary agricultural co-benefits are expressed in per capita terms. The results for the *NDC* scenario highlight the areas where ozone reductions overlap with high values of ozone-sensitive crops, such as maize, soybeans, and wheat in the US, or sugar cane and soybeans in Brazil. Whereas the *NDCs* globally raise the yields of maize, rice, soy, and wheat by 0.4-0.7%, 0.1-0.3%, 0.8-1.1%, and 0.4-0.6% in 2030, respectively, more ambitious climate policy limiting global warming to 2°C increases productivity of those crops by 0.8-1.5%, 0.2-0.8%, 1.8-2.7%, and 0.9-1.7% compared to the *Reference* in 2030 (Supplementary Information). Estimates of monetary agricultural co-benefits corresponding to yield impacts for the year 2050 exceed 10\$ per capita in some regions, when calculated using current-day value of agricultural production. Different climatic conditions across the scenarios can affect the length of the growing season and, consequently, the exposure to tropospheric ozone. This effect, not considered here, would likely raise the estimate of the benefits."

Figure 5: Ozone-related crop yield benefits due to climate policy in constant 2004-2006 dollars per capita, under the assumption of Stringent Legislation (SLE) for air pollution. Panel A (top): Difference between the Reference (REF) and the NDC scenario in 2030. Panel B (bottom): Difference between the Reference and the 2°C scenario in 2050. Valuation obtained by applying future crop productivity improvements compared to the Reference on the average gross production value of 2009-2013, and dividing by the population average of the same period.

Minor comments

* We have added references to more recent work, including:

Ou, Y., Shi, W., Smith, S. J., Ledna, C. M., West, J. J., Nolte, C. G., & Loughlin, D. H. (2018). Estimating environmental co-benefits of US low-carbon pathways using an integrated assessment model with state-level resolution. *Applied energy*, 216, 482-493.

Zhang, Y., Smith, S. J., Bowden, J. H., Adelman, Z., & West, J. J. (2017). Co-benefits of global, domestic, and sectoral greenhouse gas mitigation for US air quality and human health in 2050. *Environmental Research Letters*, 12(11), 114033.

Zhang, Y., Bowden, J. H., Adelman, Z., Naik, V., Horowitz, L. W., Smith, S. J., & West, J. J. (2016). Co-benefits of global and regional greenhouse gas mitigation for US air quality in 2050. *Atmospheric Chemistry and Physics*, 16(15), 9533-9548.

Yang, X., & Teng, F. (2018). Air quality benefit of China's mitigation target to peak its emission by 2030. *Climate Policy*, 18(1), 99-110.

Cai, W., Hui, J., Wang, C., Zheng, Y., Zhang, X., Zhang, Q., & Gong, P. (2018). The Lancet Countdown on PM_{2.5} pollution-related health impacts of China's projected carbon dioxide mitigation in the electric power generation sector under the Paris Agreement: a modelling study. *The Lancet Planetary Health*, 2(4), e151-e161.

Li, M., Zhang, D., Li, C. T., Mulvaney, K. M., Selin, N. E., & Karplus, V. J. (2018). Air quality co-benefits of carbon pricing in China. *Nature Climate Change*, 8(5), 398.

* We make reference to Vandyck et al. (2016), both in the main text and in the Supplementary Information

* We have improved Figure 2 along the lines suggested

* "“On a regional level, air quality co-benefits generally” I didn't see how the authors derive this conclusion".

The wording was indeed vague; the description of the results shown in Figure 2 has been improved:

"Figure 2 shows that air quality co-benefits generally outweigh adverse side-effects on the aggregate level for most pollutants and regions"

* The caption of Figure 3 now more clearly states "Reduction" to indicate REF-NDC/2°C.

* In the context of this study, the WHO formula for Population Attributable Fraction can be simplified, but both definitions are equivalent and consistent, which is now mentioned in the text:

"Note that the formula above is equivalent to the WHO definition (http://www.who.int/healthinfo/global_burden_disease/metrics_paf/en/) since we calculate premature mortality based on population-weighted country-level aggregate levels of concentration with the relative risk under 'ideal exposure' to air pollution equal to unity."

REVIEWERS' COMMENTS:

Reviewer #1 (Remarks to the Author):

No further comments. I appreciate the careful responses to previous comments.

Reviewer #2 (Remarks to the Author):

I reread the paper. It significantly improved. I have only a few comments:

Page 3: "This hypothetical benchmark identifies how structural changes induced by climate policy can improve air quality beyond what can be expected by end-of-pipe air pollution abatement technologies alone, and will be used here to quantify a lower bound for the co-benefits."

It took me again a while to fully understand this issue: the BAT scenario already implements stringent air pollution abatement, leaving less room for co-benefits of climate action. It would not be a bad idea to stress this point clearly at this position (even though the point is now better explained..).

Page 3...pollutants affect global mean temperature changes, as well as regional variations around the average and precipitation patterns...

This reads awkward. restructure....e.g. pollutants affect global mean temperature changes, regional variations around the average (...this refers to average changes in temperature I guess), and (regional?) precipitation patterns...

Figure 1: yellow caption on left read : INDC and not NDC

Page 3: This result does not consider the effect of air pollution controls on greenhouse gas emissions (which do not vary across air pollution control assumptions here) and is less pronounced in more ambitious climate mitigation scenarios (because co-emitted air pollutants are also reduced)

Please rephrase. Reads unpleasant and is hard to understand with the sentences in brackets.

Page 7: Different climatic conditions across the scenarios can affect the length of the growing season and, consequently, the exposure to tropospheric ozone. This effect, not considered here, would likely raise the estimate of the benefits.

I would not include this statement...."likely raise" suggest some kind of confidence, which I think is premature. Other effects, even ones not mentioned in the discussion interfere here". Think of changes in nitrogen fertilisation, "reduced" CO2 effects, enhanced respiration from agricultural soils under climate change, changes in precipitation on crop yields, etc. etc.

Reviewer #3 (Remarks to the Author):

1. Page 1 "with mean value of 49 \$(2008)/tCO₂(9)," remove "2008"

2. In Figure 1, explain "% of which CCS: 43 22 26" in the captions. It is not clear what does this mean;

Also, I suggest to use different color/shapes for BAT&FLE for plot on the left to make them discernible.

3. In supporting 1.5.1, I did not see the discussions of the projections for the baseline mortality rates and population as used the in the study.

Reviewer #2 (Remarks to the Author):

Page 3: "This hypothetical benchmark identifies how structural changes induced by climate policy can improve air quality beyond what can be expected by end-of-pipe air pollution abatement technologies alone, and will be used here to quantify a lower bound for the co-benefits."

It took me again a while to fully understand this issue: the BAT scenario already implements stringent air pollution abatement, leaving less room for co-benefits of climate action. It would not be a bad idea to stress this point clearly at this position (even though the point is now better explained..).

Done

Page 3...pollutants affect global mean temperature changes, as well as regional variations around the average and precipitation patterns...

This reads awkward. restructure.....e.g. pollutants affect global mean temperature changes, regional variations around the average (...this refers to average changes in temperature I guess), and (regional?) precipitation patterns...

We agree and have simplified this sentence

Figure 1: yellow caption on left read : INDC and not NDC

Thank you, we have made the correction

Page 3: This result does not consider the effect of air pollution controls on greenhouse gas emissions (which do not vary across air pollution control assumptions here) and is less pronounced in more ambitious climate mitigation scenarios (because co-emitted air pollutants are also reduced) Please rephrase. Reads unpleasant and is hard to understand with the sentences in brackets.

We agree and have simplified this sentence

Page 7: Different climatic conditions across the scenarios can affect the length of the growing season and, consequently, the exposure to tropospheric ozone. This effect, not considered here, would likely raise the estimate of the benefits.

I would not include this statement...."likely raise" suggest some kind of confidence, which I think is premature. Other effects, even ones not mentioned in the discussion interfere here". Think of changes in nitrogen fertilisation, "reduced" CO2 effects, enhanced respiration from agricultural soils under climate change, changes in precipitation on crop yields, etc. etc.

Fair point; we agree to leave out this statement.

Reviewer #3 (Remarks to the Author):

1. Page 1 “with mean value of 49 \$(2008)/tCO₂(9),” remove “2008”

Done

2. In Figure 1, explain "% of which CCS: 43 22 26" in the captions. It is not clear what does this mean;

Done

Also, I suggest to use different color/shapes for BAT&FLE for plot on the left to make them discernible.

Done

3. In supporting 1.5.1, I did not see the discussions of the projections for the baseline mortality rates and population as used the in the study.

You can find the reference to the relevant data source in the last lines of 1.5.1:

"the All-cause, natural Baseline Mortality (ABM) is obtained by multiplying UN forecasts of population (medium fertility scenario) with UN crude mortality rate projections (**Error! Reference source not found.**) and a scaling factor to reconcile UN data with GBD 2015"